# Modern Practice, Archaic Ritual: Catholic Exorcism in America

William S. Chavez

Department of Religious Studies, University of California, Santa Barbara, CA 93106, USA; wchavez@ucsb.edu

**Abstract:** The following ethnographic and folkloric analysis of American exorcism practices post-1998 centers on four Catholic priest-exorcists currently active in the United States. After a brief commentary regarding the place of Satanism within contemporary Catholic imagination, this article posits that the Catholic Church's recent institutional support of its office of exorcist must not be viewed separately from its discursive fear of Satanic cults and larger narratives of religious declension. The current era of exorcism practice in America is chiefly characterized as a response to the media sensationalism surrounding not only prior cases of demonic possession but also of Satanic ritual abuse. Moreover, beyond these explicit issues of religious competition (e.g., Catholics versus Satanic conspirators), the current era of exorcism practice is also implicitly characterized by the changing belief systems of contemporary Catholics. Thus, this article ultimately concerns issues related to religious modernization, the apotropaic use of established religious tradition, popular entertainment and the mediatization of contemporary exorcism cases, institutionalized training curricula and the spaces allowing ritual improvisation, and the vernacular religious consumption of unregulated paranormal concepts that possess no clear analogues within official Church theology.

**Keywords:** Catholicism; exorcism; modernization; ritual; mediatization; vernacular religion; paranormal; preternatural

## 1. Introduction

The following article expands the cultural history of exorcism presented in Michael Cuneo (2001). As such, I approach this ritual practice as a complex web of networks and ideas, accepting Cuneo's proposed dynamic of Vatican-sanctioned, rogue Catholic, and non-Catholic exorcists operating simultaneously in America. However, the following ethnography focuses exclusively on the contemporary priest-exorcists who practice in accordance with the Roman Catholic Church. I wish not to reify the Roman tradition as official or exclusively legitimate, but rather to study the religious liberties formed and available within such an established tradition.

This analysis of Catholic exorcism practices post-1998 stems from my dissertation research—interviews with dozens of contemporary practitioners, four of which are highlighted and featured below due to their informative responses, distinct exorcism networks, and specific regional contexts. None of these men were exorcists during the span of Cuneo's research, with two of them attending a training seminar in Rome unavailable to Cuneo's informants. With priority given to their responses, this article is organized thematically, with large portions of their interviews driving the structure of argumentation. With each new practitioner, I offer a short introduction regarding their history of Catholic practice before shifting to discuss how each priest contributes to our working portrait of a "modern" American exorcist.

Through my interviews with *Exorcist 1*, I introduce the ultimate stakes of the analysis: the degree to which Roman Catholic exorcism has undergone an institutional reconstruction—a modernization—such that the curriculum of ritual training now includes a space for consultations with modern health care professionals. Indeed, as Possamai and Giordan (2020, p. 1) write, exorcism should no longer be viewed as an "atavistic ritual in conflict with science and modernity".[1] *Exorcist 2*, the next featured informant, adds an interesting

caveat, demonstrating the ritual process by which the Catholic Church retains its efficacy alongside medical and psychiatric treatment. In line with the recent findings of Thomas Csordas (2017, p. 296), "[i]n addressing the relation between suffering understood to be caused by a supernatural (or preternatural) agent and suffering caused by naturally occurring illness or psychopathology", the interviews selected for this article raise "not only the issue of how scientific medicine and ritual healing interact but also . . . an abiding concern with the relation between faith and reason". In this discussion of religiously charged paraphernalia, *Exorcist 2* expands our analytic scope to include the world of the occult, at which point I offer a short commentary regarding the place of Satanism within contemporary Catholic imagination.

The Catholic Church's recent institutional support of its office of exorcist, which I shall demonstrate below, must not be viewed separately from its discursive fear of Satanic cults and larger narratives of religious declension. The current era of exorcism practice in America, I argue, is chiefly characterized as a response to the media sensationalism surrounding not only prior cases of demonic possession but also of Satanic ritual abuse. As such, the recent mediatization of *Exorcist 3* and his first major exorcism are also examined below—at which point, I share my own experience with popular media through my "Exorcism Scholar" appearance on a program for the History Channel (where *Exorcist 3* was also featured).

Through my interviews with *Exorcist 4*, we expand the scope of analysis further to include a discussion of ritual mechanics, i.e., how rites, translations of rites, and other instruments within the exorcist "toolbox" are utilized in various situations and marked as combat effective. What is important here is the freedom for each priest-exorcist to ritually improvise while in practice, how the power of the sacred is utilized differently according to the practitioner. This transition prompts, first, a discussion of "actor-network" theory, which I use to narrate the traditional Catholic exorcist's relationship to his tools, and, second, an examination of a vernacular religious dimension understudied in Americanist Catholic research, that of the paranormal. Beyond the explicit issues of religious competition (e.g., Catholics versus Satanic conspirators), the current era of exorcism practice is implicitly characterized by the changing belief systems of contemporary Catholics. The types of beings defeated through exorcism comprise the final point of concern for this article, as my informants evaluate the Catholic Church's ability to explain theologically the strange entities that continue to harass and assault its members.

Thus, the final section of this article concerns issues of (1) religious modernization, (2) the apotropaic use of established religious tradition, (3) popular entertainment and the mediatization of contemporary exorcism cases, (4) institutionalized training curricula and the spaces allowing ritual improvisation, and (5) the vernacular religious consumption of unregulated paranormal concepts that possess no clear analogues within official Church theology. Before engaging such analysis, however, I present a short section clarifying my use of the ethnographic and folkloric methodologies employed within this article—why I have chosen to structure the flow of argumentation around the responses of my informants. I then introduce the reader to the general background surrounding the current state of American Catholic exorcism, presenting a survey of five specific institutional developments that distinguish this era of practice from its predecessors. To be clear, this survey of the recent history of Catholic exorcism in the United States, following the completion of Cuneo's research, demonstrates the reasons why this material is worth revisiting a generation later.

## 2. Methodologies

The decision to prioritize informant responses within the structure of this article is inspired, first, by Leonard Primiano (1995) and, second, by David Hufford (1982). "Vernacular religion", Primiano (1995, p. 51) writes, "as an approach embracing theory and method, incorporates attention to such ongoing interpretations and negotiations of religion within groups and institutions by providing theoretical awareness and ethnographic reflexivity to

the study of such individual creations of religion". Each of the practitioners featured below offers an individual instantiation of Catholic exorcism practice and vernacular Catholicism. As such, the religious "vernaculars" of this priestly cohort (e.g., ritual improvisations, dissenting opinions, procedural preferences) are central to this article's analysis. Beyond the efforts of presenting a "thick description" of any ritual activity observed (Geertz 1973), I commit myself to the preservation of the oral "transmission" of the information gathered throughout my fieldwork (cf. Hufford 1982, p. xvi). How the content was delivered to me is just as significant as the content itself. As such, similar to Hufford's "experience-centered approach" (i.e., an empirical investigation of supernatural experience prioritizing self-initiated generic descriptions over culturally-relative signifiers), this article employs a "practitioner-centered approach". The ethnography presented in this article stems not from a select community or large survey of lay "possessionists" (those laity believing demonic possession is real) or "energumens" (those presently or formerly demonically afflicted). Instead, the information below is presented in either dialogue form between the priest-exorcist and myself or as a collection of folklore, stories that the informants share with other priests, supplicants, or parishioners. As a researcher and non-ritual specialist, it is important that I preserve and present the moments of conversation where the priests transition into familiar discourse, along with moments where my questions require an improvised answer.

In the excerpts that follow, preserved are the moments where the informants dispute or correct my interpretations; where they voice apprehension even if they agree with my assessments (e.g., "I'm not sure I've ever heard it put that way"). Despite my labeling of these individuals as "informants", as is common within the anthropology of religion, for the rest of this article, I collectively refer to these priest-exorcists as "active collaborators". Thus, the decision to prioritize their conversational materials within this article's argumentation is inspired, third, by Luke Eric Lassiter (2005, p. 17). Such a "collaborative ethnography" provides a welcoming space for the interviewees to participate beyond the traditional roles available when confronted with a research questionnaire. As such, I have sent my collaborators drafts of this article, transcripts of our interviews, and more—allowing each of them to comment not just on my analysis and assessment of the material but on the specific information conveyed by their peers. They understand the goal of piecing together a working portrait of a "modern" American exorcist. For this reason and others, I commit myself to protect the identities of those that request anonymity—especially given, as demonstrated below, the national interest a Catholic exorcist receives once publicly known.

## 3. Background

Cuneo's research covers four decades of Catholic and Charismatic practice in North America. He traces how the popular culture industry sparked a national interest in idioms of spirit possession and rituals of exorcism. As the demand for these rituals increased (post-1973), so too did the supply of various types of exorcists—each stepping in to respond to this growing phenomenon. As Cuneo (2001, p. 66) reports:

> Despite the excitement generated by William Peter Blatty and Malachi Martin, no more than one or two American Catholic dioceses at any given time since the mid-seventies had seen fit to have an officially appointed exorcist on call, and the vast majority of requests for exorcism-related help had simply gone unattended.[2]

This exorcist shortage seems to have been addressed around the fall of 1996 as "ten Catholic priests in the United States were appointed to the office of exorcist"—coinciding with Cuneo's eighteen-month research project (Cuneo 2001, p. 258). "Ten exorcists might not seem like many, especially in a country with a Catholic population of sixty million", Cuneo notes. "[B]ut American Catholicism probably has more bona fide exorcists at its disposal today than at any given time since the invention of color TV".

Much has changed since Cuneo began his research in the mid 1990s. Five events in particular make a revisiting of this material worthwhile. The first, as Cuneo (2001,

p. 265) notes, was the Vatican publication of a revised version of the Roman Catholic rite of exorcism in 1998.

> The exorcisms I attended for this research were among the last in the United States conducted according to the older rite, which had stood substantially unchanged since its publication by Pope Paul V in 1614. The new rite dispenses with most of the baroque epithets (Prince of Darkness, Accursed Dragon, and so forth) that figured prominently in the older one, and it underlines the importance of subjecting all cases of suspected demonization to intensive psychiatric examination.

The various versions and translations of this Catholic rite are a major point of concern within the next section, given that the latest version is not the only one in use. The second significant event came in 2004 when Pope John Paul II sent a letter to the Congregation for the Doctrine of the Faith (at that time headed by Cardinal Joseph Ratzinger), "mandating" every bishop to select and train an exorcist for their diocese (Josh at St. John XXIII 2016). Two further letters (c. 2010, 2018) would be sent in the years following, reiterating John Paul II's mandate. The latter of these, according to one of my collaborators, was sent by Cardinal Robert Sarah, former prefect of the Congregation of Divine Worship and the Discipline of the Sacraments (CDWDS), and stated "Every diocese must not only have one but you are to supply me the name of that priest". The third significant event, also in 2004, was that the school of theology at Regina Apostolorum, one of Rome's most prestigious pontifical universities, devised a special program to not just educate its students on Satanism but also to train a new class of exorcists (Day 2005).[3] This event is perhaps the most crucial to the current study. Issues of standardizing practice and discernment, religious declension, media sensationalism, and degrees of esotericism all contribute to the significance of this Roman Catholic course.

The fourth and fifth events then go hand in hand: a rise in the number of American priest-exorcists and an organizational meeting sponsored by the United States Conference of Catholic Bishops. After the training program was initiated in 2005, multiple media outlets reported a rise in demonomania and the number of ritual specialists. In 2009, Msgr. Gregory Ketcham, the director and head chaplain at the St. John's Catholic Newman Center (Illinois), said that there were only 12 exorcists in the United States (The Daily Illini 2009). However, reports say that 56 bishops and 66 priests registered for the Conference on the Liturgical and Pastoral Practice of Exorcism (in Baltimore) the following year (Sadowski 2010).[4] According to one of my collaborators who attended the event:

> It was kind of a pre-conference workshop, for tribunal folks, just in processing and handling their cases. More a very preliminary series of talks, just to help tribunal staffs. Just to establish protocols. How do you process questions? All of those things.[5]

Shortly after the conference, one exorcist, Fr. Gary Thomas, estimated that there were approximately two dozen trained specialists among the 185 Catholic dioceses in the country (Otis 2010). In my 2018 interview with Fr. Thomas, he estimated that the number is now closer to 150.[6] Fr. Vince Lampert, exorcist for the Archdiocese of Indianapolis and of the same exorcist cohort as Thomas, said in 2019: "I'd say there are at least 175—and more each year" (Hopfensperger 2019).[7]

In short, though unfortunately coinciding with the completion of Cuneo's primary data collection in 1998, the Catholic Church began revamping its office of exorcist with significant institutional support, leading to multiple training programs, papal mandates, revised ritual manuals, and an increased number of Catholic priest-exorcists. This is all the more intriguing to learn given that, contrary to public perception, exorcism is not an exclusively Catholic enterprise. Based on estimates of the number of practitioners (and the size of their clientele), Protestants—especially Evangelicals and Neo-Pentecostals—actually dominate the demon-busting market. However, as a religious practice, the Catholic performance of exorcism is continually popularized through American film and television.[8]

At the same time as this Catholic reconstruction of its office of exorcist, the ritual has slowly built momentum in American cinemas. This concurrent phenomenon has produced over thirty exorcism films since 1998, including the two prequels to the 1973 film *The Exorcist* (in 2004, 2005), *The Exorcism of Emily Rose* (2005), *The Rite* (2011), *The Devil Inside* (2012), *Deliver Us From Evil* (2014), and *The Vatican Tapes* (2015). Thus, although the study of exorcism in the United States can in no way be limited to the dealings of traditional Roman Catholics, Catholic exorcism is particularly appetizing for sensational media outlets and popular entertainment industries due to an acquired taste.

## 4. American Exorcism Revisited

### 4.1. Exorcist 1: Fr. Gary Thomas (White Male, Boomer, California)

In the summer of 2005, this San Francisco native and former mortician traveled to Rome as a priest for the newly designed training seminar on exorcism. After the course was complete, Thomas entered into an apprenticeship with Fr. Carmine de Filippis, a senior Italian exorcist. This portion of Thomas' life was published in a book in 2009 and adapted into a film in 2011. For the latter, he spent a week on set as a technical consultant, advising the actors for a single exorcism sequence. He later participated in several media junkets promoting the film's "authenticity". Since returning from Rome, he has served as the official exorcist for the Diocese of San Jose and now travels the country to educate the American public on the reality of supernatural evil and the need for exorcism to be taken seriously.[9]

Fr. Thomas is perhaps the most famous Catholic exorcist in the country; he is an institutional fixture and not just a media focus. Thomas, for instance, has spoken many times at the Pope Leo XIII Institute (Mundelein Seminary in Illinois), served as a board member for a number of years, and is also a member of the International Association of Exorcists. This latter organization was founded in 1990 and received formal Vatican recognition in 2014 (Glatz 2014). The former was founded in 2012 and has included the likes of experienced exorcists Msgr. John Esseff (formerly of the Diocese of Scranton, PA), Fr. Vince Lampert (Archdiocese of Indianapolis, IN), and exorcism-sympathizer Bishop Thomas Paprocki (Diocese of Springfield, IL) as long-contributing members (Armstrong 2017). While the institute's first conference convened in 2005, it was not until 2010 that the board was "approached with a request to establish a specialized and intensive educational program for priests who work as exorcists and for the deacons who support them" (Pope Leo XIII Institute 2021). The institute's two-year cycle of exorcism curricula would then produce its first cohort of graduates in 2014. Their website currently lists 163 graduates (priests and deacons) in total.

In the interview excerpt below, Thomas shares his opinions of the two exorcism training programs (in Rome and Mundelein), how their teaching cycles and pedagogic concerns distinguish one from the other.

> **Chavez:** How does Mundelein compare to the training you went through in Rome?
>
> **Thomas:** There're strengths and weaknesses in both. In Rome, it's all taught in Italian. [ ... ] I had a translator ... and in those days, it was taught every four months on Thursdays. Now they got a clue that, hey, if you want people to come on this thing, you've got to be able to teach it in a week. So it's eight hours a day for a week as opposed to four hours a day taught over four months.
>
> **Chavez:** What's the Pope Leo XIII structure?
>
> **Thomas:** Well, the Pope Leo XIII is taught for 10-day periods, twice a year, for two years. [U]sually you have 50 priests in the class. [One] 10-day session in February. [Another] 10-day session in November. Twice. Then you get a certificate. The weakness of that program or that training [is that] there's no practicum. In Rome, well, when I went there, there wasn't. I had to go find it. I had to go find Carmine. And that was not easy to do. [But] I worked it out. [I worked] for three-and-a-half

months, three days a week, for three-and-a-half hours at a shot. I saw a lot of exorcisms. And honestly, that's kind of how I learned to do . . .

**Chavez:** . . . the practicum?

**Thomas:** Oh, yeah! That's how I learned to recognize the signs (no. 1) and how to do the rite (no. 2). And, you know, to realize that there are these manifestations [that] can be ferocious. Now, in Italy, they don't do anything like what we do. They're all a bunch of lone rangers. Here, like, I have a team. I worked hard to have a team. [Or rather] I have two teams. I have a medical doctor. I have a clinical psychologist/psychiatrist. Actually, I have two psychiatrists. One's bilingual. And then I have a prayer team and the prayer team was with me at every exorcism and every deliverance. The professionals are not but they're part of the discernment. Because when people come in . . . the team now usually handles the interviews. I don't really. I'm not involved with the interviews like I once was because I got too much going on. And we can have anywhere from five to a dozen people [that] we're praying over every single month. It takes a hell of a lot of time. Their time too, you know? I get lots of calls and they start off with, "I need an exorcism". And my pet answer back is, "I don't do them on demand". And that may not be what they need or what they wanna hear. Sometimes they're pleased and sometimes they're not. But like I say, "I don't do them on demand". It doesn't work that way. And this [exorcism] may not be what you need. We'll have them evaluated. Usually we use a clinical psychologist rather than a psychiatrist. The psychiatrist comes into play if they've been under the care of a psychiatrist and they're on their meds. Only reason to send for the psychiatrist is to say, "Okay, are they on the right meds and are they on the right dosage?" And, you know, "Is what they're describing [legitimate]?" "What do you think in light of what they're sharing with us?" And [we say], "Go have the interview. See what you think". And then we go from there.

Catholic exorcists working close with modern health care professionals is a recent development. Thomas serves the Catholic Church in America as a representative to the public for what to expect from a "modern exorcist"—an epithet featured in the subtitle of the book covering his time in Rome (see Vice (2016) and Cox (2016)). He is cautious as a practitioner yet unwavering in his belief in Satan as a supernatural entity. These qualities are significant because, for the last twenty years (Watanabe 2000), exorcism news stories have sensationalized a recurring set of issues specific to the practice, including the increasing use of the "medieval" or "secret" ritual (and its place in modernity), exotic autodidactic methods (not just holy water and scripture), monetary issues (what exorcists charge for their services), botched performances and scandals (involving assault, rape, and/or murder), and controversial clients (homosexuals, for instance). Priests such as Thomas then offer scholars a "modern" portrait of an "archaic" ritual practice, how contemporary exorcists are currently "modernizing" one of Christianity's oldest faculties. By this, I imply no support for the cultural evolution narrative, à la Tylor (1871, pp. 15–16), whereby exorcism persists in modern society as a "survival" soon to be discontinued as the public becomes more "rational" or "advanced". On the contrary, despite the popular perception that exorcism does not belong in modern society, the practice persists and, therefore, should be studied—particularly its methods of modernization, such as the appropriation of scientific and/or therapeutic rhetoric, development of new theologies, use of new technologies, promotion across various social media platforms, and consultations with the production of exorcism films and television series.

### 4.2. Exorcist 2: "Fr. Barron" (White Male, Boomer, Southwest)

Like Fr. Thomas, this next collaborator received pontifical training in Rome (under its original two-part seminar curriculum), became a member of the International Association of Exorcists, and has served his diocese as exorcist for over ten years. This priest, however,

entered into an apprenticeship with an American exorcist stationed in Idaho—a process he believes is the best way to learn the craft. Before proceeding with the ritual, this priest seeks to disqualify a host of alternative conditions to possession, such as Tourette's syndrome, sleep paralysis, paranoia, delusion, and the like. While psychological factors, he says, such as trauma and resentment, can serve as "open doors" for supernatural predators, he believes that we as a society have forgotten about the *spiritual* dimensions to demonic affliction. People can suffer, for instance, because "they do not know how God can love them". Yet, because of this "hermeneutic shift" when assessing the human condition, we now tend to favor the more medical and psychological systems of evaluation, "Fr. Barron" explains.

> What I'm talking about is not the use of psychology within the context of sacramental worldview but the abandonment of the world of the spiritual. And the replacing of it with psychology. I think it starts early in the 1960s. When I was in grade school, everything was talked about from a spiritual aspect. When I went into high school in '65, they had begun to let go of that idea. And they began to talk about the psychological. So instead of *holiness*, it was *healthiness*. We want people who are *healthy*. Now what determines health in this world? Your ability to relate to people, your flexibility, politeness. And *sin* was soon replaced by *illness*. Now, that continues from the time I entered high school [and into] my studies at the university and later seminary. In my priest group, you lived with eight or nine guys and then you prayed with them and you talk to them. But it was always about the psychological. The focus was always on "How well integrated are you in here?" "Is there something in you which keeps you from integration?" "Is there something in you that will prevent you from being a leader in the community?" "Is there something in you, something missing, something lacking, something skewed, that would not allow you to deal well with people once you're ordained?" Those are all the questions they asked. Never said, "Is he praying?" "Going to church?"[10]

Holiness is key to salvation, according to "Fr. Barron"—and he indeed weaponizes it. During the diagnostic stages of his work, he tests for an aversion to anything sacred on the part of the individual. He offers them Catholic medals, blessed beforehand, and observes their response. He invites them to sit first at a table, before which he has traced a cross in holy water or blessed salt on all but one of the seats. During our interview, I inquired how, given that most of his exorcized subjects are non-Catholic, does he then treat those of another faith or creed.

> **Chavez:** Are there ever questions of conversion?

> **"Barron":** Never, never. The reason I don't bring it up is because that's not the issue. The issue is the question of their spiritual health. If they're Baptist and not practicing then I will give them hell for not being a practicing Baptist. But my answer is "You need to practice your faith". Whether it is Baptist or Methodist or Lutheran or Assembly of God. [ . . . ] We often get people who say that they have a spiritual issue and they go to their minister and their minister says, "No, this doesn't happen", or "This ended in the Apostolic Age". Some ministers will go even further. They'll say: "No, no. The problem is you don't have a deep enough faith". And some ministers say, in an a really venal sense, "If you contributed more, this would not happen". So then they will bounce around going from minister to minister and generally somebody who is in connection with them who is Catholic will say, "You need to see a priest. They know how to deal with this". So then they'll come to me and say, "Can I talk to you about this? And I don't know how to describe it. I don't know what to say but I got a problem". And that's the point at which I talk to them. But no, it's never dealt with the question of conversion.[11]

As our conversation continued, it quickly became apparent that "Fr. Barron" is quite a researcher himself. Using the information gathered from those possessed by the occult, he created a database of temporal observations, a document entitled "Witchcraft and Satanic Cult Calendars, For the Year 2018–2019".[12] The document cross-references the dates of the Gregorian calendar with the occult feast days of numerous pagan systems, including Celtic, Norse, Greek, Roman, Canaanite, and Egyptian. The document also lists the corresponding Catholic feast days, lunar phases, and the Satanic meanings behind each date (most of which involve blood and/or sexual types of sacrifice). Within this religious cosmology, a great power resides within the emblems and rituals of Catholic tradition and the same appears true for the occult. However, to what extent?

> **Chavez:** What counts as the *occult*? Is it items or symbols of other religions that are non-Catholic or non-Christian? A Hindu om or a Muslim insignia? What about non-Catholic practices like yoga and meditation?
>
> **"Barron":** Somebody who's a Muslim, with the elements of their faith, somebody who is Buddhist or Daoist with the symbols of their faith, those are not demonic. Yoga was originally conceived as a way of entering into communion with a particular god. So different positions and different mantras were about raising your level of consciousness into a field of divinity in which case, by assuming a posture, repeating or breathing a certain way, you would achieve union with a particular god. [ . . . ] For most Americans, it's a way of stretching and limbering and even those people who are very strong devotees of yoga are not going to be aware of the spiritual element to it. Is that occult? No, I don't believe so.[13]

The above sentiment from "Fr. Barron" then reflects the larger American cultural idealism surrounding religious pluralism. He prioritizes faith and religious practice above sectarianism as both directly correspond to issues of "spiritual health". At the same time, occult practices serve as forms of ritual abuse and must be taken seriously by the Catholic Church, he affirms. The work of the "modern exorcist" then involves bringing people closer to divinity and ridding the world of its supernatural evils through a careful study of the "Enemy" and his followers.

### *4.3. Exorcist 3: Fr. Michael Maginot (White Male, Boomer, Indiana)*

While the previous priests serve as fully appointed diocesan exorcists, this next collaborator only works on a case-by-case basis—and has done so since his high-profile first case in 2012. Maginot was also never formally trained at the Regina Apostolorum in Rome, Pope Leo XIII Institute in Mundelein, nor within any sort of apprenticeship. Instead, upon initially receiving permission from his bishop, he conversed with other experienced exorcists in the nearby area.[14] Today, he assists in cases as much as the bishops allow, sometimes even with those outside of his Diocese of Gary.

Like "Fr. Barron", Fr. Maginot is cautious about the dangers posed by dark supernaturalism. His view of what constitutes the "occult", however, is far more expansive.

> **Chavez:** What are the common reasons for how people become supernaturally afflicted? Is it a crisis of faith? A haunted house? A rough time?
>
> **Maginot:** Some could be the place, the location, that they are in a house that had some things. That's kind of a common thing. The second thing, they could have been delving or playing with the occult. [ . . . ] Did they have their palm read, or tarot cards, or played a Ouija board as a kid? If they say "No, never did anything like that", then the next thing is, did they ever have a relationship with someone who was kind of into those things? Trying to get them into it? Did they ever give you something or did you ever give them something? If so, [then] they got a cursed object – which happens a lot. That's a good thing, actually; you don't need an exorcism, just getting rid of the cursed object is fine. [ . . . ]
>
> **Chavez:** What constitutes a cursed object?

> **Maginot:** I would say it has to be something where a demon would want to be attached to it to get at a person. Something that won't be thrown away. [ . . . ] It's usually something like jewelry, or a necklace, or something that's kind of valuable that, you know, seems precious and not necessarily has demonic signs or anything. It may just be some sort of gem type of thing, you know? [ . . . ] A lot of it could also be they went to an antique shop, saw this thing at a garage sale. A lot of times they don't remember that. So I have to say, "Did you ever go to antique shop?" And so it is: "There was this cute little thing". It's usually something that just catches their eyes, and then they kind of get enamored and drawn to, you know, but then it's hard again to get rid of these attachments.[15]

Another of my collaborators shared a related story about an occult object he still keeps in his garage. It originally belonged to a woman in his parish who visited a New Age bookstore. While the original figurine bore a likeness to a fairy, after the woman cited strange domestic occurrences, the priest instructed her to remove all New Age materials from the house. As the family was moving the bagged items to the trash, the figurine broke. "Embossed in the figure was a Satanic altar", the exorcist says, "with a body, a statue, of a priest headless". "His head glued to the altar, the body glued to the altar, the hands glued to the altar. I mean, the creepiest thing I ever saw".

Recall that the Roman Catholic course on exorcism includes a survey of Satanic practices within its curriculum—and has done so since its inception. Satanism is also a perpetual concern for both the International Association of Exorcists and its late co-founder Gabriele Amorth, making the subject a clear institutional focus on the part of the Church (Martin 2019; Wright 2017).[16] Such discourse along with the folklore of these exorcists reveal that we still live within the aftermath of the Satanic Panic (1980–1993), a popular media scare that swept through North America and Europe detailing the existence of an underground network centered on Satanic ritual abuse.[17]

In 1992, Kenneth Lanning, a Supervisory Special Agent with the FBI's Behavioral Science Unit in Quantico, Virginia, authored a report for the Department of Justice, challenging the criminal significance of occult symbolism and the assumed narratives generated by the public. As Lanning (1992, p. 45) concludes:

> Overzealous intervenors must accept the fact that some of their well-intentioned activity is contaminating and damaging the prosecutive potential of the cases where criminal acts did occur. We must all (i.e., the media, churches, therapists, victim advocates, law enforcement, and the general public) ask ourselves if we have created an environment where victims are rewarded, listened to, comforted, and forgiven in direct proportion to the severity of their abuse. Are we encouraging needy or traumatized individuals to tell more and more outrageous tales of their victimization? Are we making up for centuries of denial by now blindly accepting any allegation of child abuse no matter how absurd or unlikely? Are we increasing the likelihood that rebellious, antisocial, or attention-seeking individuals will gravitate toward "satanism" by publicizing it and overreacting to it? The overreaction to the problem can be worse than the problem.

As it turns out, despite the moral panic, media hysteria, and criminal accusations, there was never any evidence to suggest the existence of a systematic underworld of Satanic ritual abusers (though many exorcism ministries and practices emerged during this time to confront them).[18] That said, I do not mean to imply that my collaborators and others are obsessing over some purely imaginary set of villainous pagans.[19] In 2018, for instance, many Catholics voiced collective outrage upon learning that a coven of witches in Brooklyn, New York had organized an event that would hex and curse "all rapists and the patriarchy which emboldens, rewards, and protects them"—the most notable offender being Supreme Court Justice Brett Kavanaugh, a pro-life Catholic. $10 tickets were sold to the event, with half the proceeds going to women's and LGBT charities, as the same group held similar ceremonies in the past to hex Donald Trump and others (Wolfson 2018).

Fr. Gary Thomas condemned the group and event in several statements made to a Catholic periodical, saying:

> I'm appalled. I sent this to a load of exorcists yesterday and their reaction was similar to mine. That shows this is not something that is make believe. [ ... ] Conjuring up personified evil does not fall under free speech. Satanic cults often commit crimes; they murder and sexually abuse everyone in their cult (Armstrong 2018).

Thomas also offered two Masses for Kavanaugh, one of which was on the same day the coven planned their hex ceremony. The Diocese of San Jose would later release a statement on their website clarifying Thomas' comments and actions:

> Father Thomas said that he would offer the regularly scheduled Mass on both dates for the Justice and that he would pray for the safety of the Justice during each Mass's Prayer of the Faithful. In addition, Father Thomas never said that the gathering was being led by a Satanic cult. He made comments about Satanic cults, but those comments don't apply to the group placing the hex on Justice Kavanaugh because this group refers to themselves as a coven, not a cult (Gafni 2018).

Thus, for many of my collaborators, the "work of Satan" refers simultaneously to: (1) the supernatural beings that tempt, oppress, and possess Christians and non-Christians; (2) the occult world that may sinisterly reside in common, innocent-looking household objects; and, (3) those beings of flesh that, intentionally or not, do the Devil's bidding. In 2011, in an article for *U.S. Catholic*, Nancy Caciola, a medievalist scholar and author of the "Exorcism" entry in the *Encyclopedia of Religion* (2005), argued that, generally, "exorcisms are associated with these turning-point moments when the church [feels] challenged in some way and tries to centralize power and clarify the delegation of authority from God down through the hierarchy" (Burke 2011). The article then lists multiple challenges confronting the Catholic Church in the United States, including "the sex abuse scandal, a secularizing society, and a restive flock that, studies show, loses one out of three adult Catholics, to name just a few" (see Caciola (2003, 2005)).

My study then demonstrates that the challenge of Satanism should be added to this list; the Satanic Panic represents a clear period in Christian history that has produced not only discursive fear but institutional support for one of the Church's oldest offices. Giordan and Possamai (2018, p. 11) likewise argue that exorcism "peak[s] in the public sphere and in people's everyday life consciousness" when scientific discourse is weakly emphasized and at times of social crisis (specifically when religious groups are competing against each other).[20] However, given that more and more exorcists are employing medical and psychological evaluations of their subjects, the authors clarify that the popularity of exorcism "is not due to an increase in superstition, but to religious market forces. In this sense, exorcism is now a commodity object used to brand certain religions" (p. 96). Satanism, especially today, is seen by many of my collaborators as a legitimate threat to Christianity. As Fr. Thomas said in our 2018 interview:

> Is everything demonic? No, but Satan is a real, is an intelligent being. The scriptures? That is not metaphorical. It's not. And I'll tell you, there's plenty of Satanic cults around the country. They get a big kick out of this, when people think it's all made up because it just continues to let them flourish. I mean, Satanic cults are criminal. They perform crimes against humanity. They murder. They're secret and there's loads of people. And it doesn't just run in cultures *per se*. There are lots of lots of people. I get interviewed and asked, "Why is the church making so much more out of this now?" I says, "Because our country's becoming pagan". And it is! Benedict XVI said, "As faith diminishes, superstition increases". And that's exactly what we're seeing, as attendance in all churches, but certainly in Christianity, declines. We are in a major, major slide. Catholic worship in the United States is about 22 percent. And honestly, I don't know

where we're going to be in 10 years because the Millennials, only 10 percent even consider themselves connected to any church. So what's going to happen to all of our churches and our houses of worship in the next 10 years? It's going to be very different.[21]

Cases of demonic possession and allegations of Satanic ritual abuse continue to serve sensationalist media outlets well, as the American public tends to possess a renewable interest in these topics. Catholic priests such as Gary Thomas and others are then highly-sought figures in an attempt to further mediatize the office of exorcist into a socio-religious drama staged before the nation. Fr. Maginot was one such individual, launched into the national spotlight once he served as the exorcist in the Latoya Ammons case in 2012. The story became a news sensation in late January 2014 (Kwiatkowski 2014). By February, the haunted home was purchased by *Ghost Adventures* host Zak Bagans (Huffpost 2014). The house was eventually demolished in 2016 but not before Bagans filmed *Demon House* (2018), his documentary on the subject (Golgowski 2016). Additionally, the media company Relativity—that had previously released other paranormal horror films such as Oculus (2013) and *The Lazarus Effect* (2015)—acquired the life rights to the Ammons family in 2014, only for the company to declare bankruptcy the following year (Kit 2014; Patten and Fleming 2015). While no film has been made, the Ammons case was featured prominently on the History Channel (see (Bizarre Rituals 2019)). In an episode of *The UnXplained* series (2019), separate interviews of both Fr. Maginot and myself were featured. When we met afterward, the two of us quickly bonded over an appreciation of exorcism cinema, confirming my intuition that priest-exorcists are quite familiar with the mediatization of this religious practice.

> **Chavez:** Do you enjoy the movies like *The Rite* and *The Conjuring*?
>
> **Maginot:** I do like, actually. The real stories! You know, the ghost things, I think if you've seen one you've seen them all. But I always seem to get into the demonic ones.
>
> **Chavez:** Are you familiar with Ed and Lorraine Warren and *The Amityville Horror*?
>
> **Maginot:** *The Demonologist*, right? Was that their book? I think that was around the time of *The Exorcist* too. I kinda like the Warrens, but it seemed like they were out on an island at the time. They were like Lewis and Clarke. It seems like they were always trying to get priests to help with their calls.
>
> **Chavez:** They actually sold the rights to their cases and characters; now they have actors playing "Ed and Lorraine Warren" in the movies. And now that Lorraine is dead, Tony Spera, her son-in-law, runs their paranormal team and he's been rather outspoken at how Hollywood the stories have now become, like the *Annabelle* movies and *La Llorona*.
>
> **Maginot:** You know, it was kind of a battle because of Tony DeRosa-Grund. He's an executive producer and he had the rights to the Warrens, but he would always get in conflict and you know, battle everyone on that. And I think he was gonna do the same to me. [ . . . ] I know because I also worked out with Zak Bagans, because I was going to do the *Demon House* documentary. And I was kind of playing both against each other. I was giving one rights to do the real story, with the real people, the real place, the documentary, and then [DeRosa-Grund] could do the movie thing, and you know, have actors and soundstages and all that. But I wanted kind of both to be real. And so that was kind of his work to tell the real story in the movie, because I don't want a priest being thrown around like, all over, you know? Dramatic things that didn't happen! So that was what kind of sold me on the project. But then he got into all kinds of lawsuits and everything, and now Hollywood never wanted to deal with him, and he is suing everyone and Warner Brothers. He thinks, you know, that if he pays you a dollar that he has you for life and it doesn't quite work like that. So I think he was

doing the same thing with the Warrens; he was kind of doing that with me. [ . . . ] Sony-Colombia was also very much interested, but they wanted to do it with Latoya was well—which was good.

**Chavez:** For her to sell her rights and use her name? Because she doesn't show up in the Zak Bagans' documentary.

**Maginot:** No, she refused. There is a boyfriend she's involved with now, that's like her agent. And he makes it like, you know, he's going to tell his story. He wasn't even in the picture at that time! He got Relativity involved. They took her story and Hollywoodized it, and they went bankrupt over all that. So we've never been able to get together to tell the true story.[22]

*The Exorcist* (1973) featured multiple priests serving as technical advisors on set, including Thomas Bermingham, William O'Malley, and John Nicola. Of the films released after 1998, Fr. Gary Thomas served as a consultant on *The Rite* (2011), Evangelical exorcist Bob Larson on *The Devil Inside* (2012), demonologist/medium Lorraine Warren on *The Conjuring* (2013), and NYPD sergeant turned Catholic demonologist Ralph Sarchie on *Deliver Us From Evil* (2014). Additionally, the story of Fr. Vince Lampert, exorcist for the Archdiocese of Indianapolis, was the subject of an episode of *Paranormal Witness*, a British documentary series on SYFY (see (The Exorcist 2013)). Perhaps someday we will see a full-length, dramatic reenactment of Fr. Michael Maginot's story. Regardless, this recent trend shows what Douglas Cowan (2008, p. 183) identifies as "the sociological power of the 'true story.'" Many exorcism films involve some variant of the coveted marketing gimmick "Inspired by True Events" or utilize found-footage techniques to thematically buttress the subject matter (see Olson and Reinhard (2017), pp. 125–47). As we saw with Fr. Thomas, these recent consultants/subjects then promote the film's "authenticity" during their respective press tours. As Bob Larson said of his film, "*The Devil Inside* is the closest that Hollywood has come to getting it right" (Movie Trailers 2012).

When it came time for my interview with the History Channel, I soon realized that I was chosen due to my proximity to Los Angeles along with my publicly available master's thesis (Chavez 2018). The interview took approximately three hours as the producers were repeatedly, albeit politely, disappointed with my answers to their questions. The role they incessantly tried to bestow upon me was of an educated, former skeptic who now confirms the validity of supernatural evil and the efficacy of the ritual of exorcism to an equally-skeptical American audience. As if to say, "I was once like you, TV viewer. But then I started studying exorcism and now I know that it's real". "Please talk about those incredible things that possessed people do", they asked in reference to Latoya Ammons' son reportedly walking backwards up a wall. "How can we rationally explain these things? Does this prove demons or demonic entities exist?" My personal beliefs were neither the subject of that interview nor this article. As a result, my place in the episode was reduced to less than 30 s. My segments added nothing of academic value, providing only narration of the Ammons case.

One question I was repeatedly asked by the producers—as an "Exorcism Scholar"—was why "the demons possessing Latoya were more responsive to the Latin prayers than the English ones". What they wanted from me was a confirmation that the demons know the power of God when they hear it—which can be found within my collaborator interviews. There is a consensus formed around this use of the Latin rite over its English translation, as Fr. Thomas explains in the excerpt below.

**Thomas:** When I was first exorcist, I was using this English translation that wasn't approved. But I did not know that, and because [the former exorcist of Chicago] said you could use it when in fact you absolutely could not. So then I started using the Latin. And fortunately, I really only had one or two cases where I used this bogus English translation on people when I shouldn't have.

**Chavez:** So when you were at the pontifical university . . .

> **Thomas:** . . . at the [Pontifical Athenaeum] Regina Apostolorum . . .
>
> **Chavez:** . . . they gave you a Latin or Italian version? Not the English one?
>
> **Thomas:** It wasn't approved then. The approved English is only a year old [c. 2017]. I know one of the priests who worked on the translation into English. He's a friend of mine. But I had the whole rite translated into English by somebody who was at the referendum college—just for my personal use. I never used it! But I had four years of Latin in high school. I studied the Aeneid and Cicero but, you know . . . I know some words in Latin but not enough in order to really understand what are these prayers really saying in English. So I had this guy translate the whole thing for me. I paid him. But I never used it. I just used it for my personal reflection per se and then it sat on the desk of the Prefect for like four or five years before it was actually approved after the bishops had voted on it.[23]

In 2016, Thomas gave a talk organized by a California group called Catholics@Work, where he reportedly shared that "The Devil hates Latin, it is the universal language of the Church". When asked afterward to clarify, Thomas said that "he had heard from exorcists who did exorcisms in Italian, Spanish and Portuguese (the only approved vernaculars for this Rite) that Latin was the most effective language" (Clayton 2016). Latin then carries with it the history of the Catholic Church as leverage against supernatural evil while English, even when there were translations available, was unsanctioned until only recently. The other English translation Thomas referenced above—recommended to him by the former exorcist of Chicago—was contained within the *Rituale Romanum* series translated by Fr. Philip T. Weller (1948–1952). Naturally, this translation was of the 1614 rite which is still the preferred weapon of choice among certain priest-exorcists.

"Fr. Barron", for instance, although he also prefers the 1952 Weller translation of the rite, exclusively uses the Latin from 1614. As he explains below, this is mainly due to the 1998 revision altering the type of *rhetoric* used in the 1614 rite.

> **Chavez:** So even though there is now an approved English version, the Latin is what you use?
>
> **"Barron":** That is what many exorcists use, only the Latin of the old rite.
>
> **Chavez:** Is there a reason why you don't use the revised version?
>
> **"Barron":** Yes. The revised version has reworked the exorcism into a liturgical ceremony. And, as a liturgical ceremony, it doesn't quite work because you're dealing with a demon that despises anything liturgical. As soon as you start praying, the demon becomes generally infuriated, and the idea that you would have someone in the family start doing the first reading and the second reading, it's assuming that the demon is going to sit there like a Presbyterian old lady and it's just not going to do that. That's my difficulty with it. The other thing, I think, is that when you become used to the old ritual, it has a certain advantage. It starts with the litany of the saints which is particularly helpful in that it allows you to recognize how the demon is reacting, what's going on, how easily provoked it will be.
>
> **Chavez:** Do you believe it was a misstep by the church to revise the rite as a liturgical ceremony?
>
> **"Barron":** I'm not sure that I'd say it was a mistake. I'd probably be more likely to say that it wasn't thought-out well enough. I mean, [the revision] was certainly done by liturgists and not by exorcists.
>
> **Chavez:** Was that part of the Church's goal? To clean up the ritual? Take out the diatribes so that it's more about praising God than about damning Satan?
>
> **"Barron":** I'm not sure I've ever heard it put that way. There certainly is that; they certainly wanted to clean it up, yeah, and put it in a way that was more

> contemporary. But I've never heard somebody say that their intention was to take out those deprecations. They certainly modified all of that, but they eventually put some of them back in as the appendix. Still, I don't think that the ritual is as strong as it needs to be.
>
> **Chavez:** Do you find that those diatribes in Latin are useful?
>
> **"Barron":** Oh, yes. It's always useful.[24]

To reiterate a point made earlier, this entire dynamic of which rite to use and what to do with the English translations is not something discussed within Cuneo's cultural history. For these exorcists to voice apprehension regarding the revised rite is significant as it demonstrates agency and dissent on the part of the practitioner. Despite the office of exorcist receiving much more institutional support than previous generations, it seems clear that the top Catholics making decisions are not exorcists themselves and may not understand the ritual mechanics involved when performing an exorcism.

My interview with Fr. Maginot reveals not only some of the tactics governing the switch between English and Latin versions of the rite but also the ritual initiative required of ad casum exorcists (those serving on a case-by-case basis) (cf. Béliveau (2020, p. 79)).

> **Maginot:** I have done 15 sanctioned [exorcisms]. Just always got permission, and just kind of learned by experience. But I did talk to [fellow priests about] the ritual. But at that time, the new ritual was in the process of being revised, and I'd heard a lot of complaints about it, so I went with the old. I guess they had permission to do the new ritual in English and Latin. But since there was so much complaints about the new, I said, "oh, I'll go with the old".
>
> **Chavez:** What version or translation? Do you use it when it's in English?
>
> **Maginot:** It was the one that was available on the Internet. They told me to go on the Internet and I got the English and later got the Latin too. [ . . . ]
>
> **Chavez:** About how long does it take either per session or for a sequence of sessions for a person?
>
> **Maginot:** You know, when I had the strength, I could go probably four hours. And that's going through the rite three to four times. I always like to start out in English. I find Latin more powerful but you don't want to get it out of control, you know? So I kind of see where you're at baseline, with English. And if you're getting a lot of manifestation, I'll stay with the English. And if you get hardly anything with the English, then I go to the Latin. [ . . . ] I do talk it over with the person, how they feel and everything, and if it's been like two hours, you know, I would probably continue if they still feel they have something. In the early years, I would try to go, you know, another two at most. You know, but I don't think I ever go beyond four hours now. [ . . . ] I also want to make sure I could complete [the rite]. I don't want to stop, you know, in the middle. So I'll have to see how much energy I have [before starting another recitation].[25]

Let the reader understand that the various translations and versions available to a priest today reflect the current period of Roman Catholic exorcism. In fact, after the 1998 revision, many of the exorcists in Rome would request an *indult* (permissible deviation from Church law) to continue using the 1614 rite—which can still be done today.[26] The subsequent 2004 edition featured a few emendations in an attempt to appease this dissenting group of priest-exorcists. Additionally, as Fr. Maginot shared, although there was another English translation available online (c. 2012), after the 2017 edition was completed, the official rite is now only given to priests directly by their bishops. Fr. Thomas views this decision as a recent effort by the Church to thwart the many Satanic cultists that were previously purchasing the ritual as a means of reconnaissance, thereby programming themselves to resist the triggering rhetoric of the opposition's chief weapon.[27]

*4.4. Exorcist 4: "Fr. Drexel" (White Male, Boomer, Midwest)*

Unlike *Exorcists 1* and *2*, this collaborator is not the product of the recent exorcism training seminars offered at Rome or Mundelein. Instead, this exorcist's path consisted of doctoral research on the subject beginning in 1998 and a three-year apprenticeship with an experienced practitioner. He is now a well-respected figure within the Catholic Church, invited to teach around the country, in the past including the Pope Leo XIII Institute as well as a series of trainings within a Texas diocese alongside Dr. Richard Gallagher, an Ivy League-educated, board-certified psychiatrist who assists the Church in discerning legitimate cases of possession (see Gallagher (2016, 2020)).

> **Chavez:** I hear a lot about exorcists being the trained skeptics. Do you think it was that way with your predecessor? Or do you find it's the same in, say, Italy or Europe? Because when you read Gabriele Amorth, he's on a different side of the skeptic spectrum.
>
> **"Drexel":** Very much. And maybe it's part of our American culture in a sense because we are so medically based—if you will. In a sense of diagnosing and discerning. But, I guess for me, I'd rather err on the side of . . .
>
> **Chavez:** . . . caution?
>
> **"Drexel":** Well, that as well. But I mean to treat the *whole* person. I think the Church has always exercised caution. You go back and read the *praenotanda*, the guidelines to the 1614 rite. In the sense of ruling out [alternatives], they talk about an equivalent to what would be a type of depression. They use the word "melancholy". And so even then, long before medicine was so formalized, for the Church to say "Well, test the waters" is significant. I think what governs all of this is the Holy Spirit, is a matter of discernment. What is going on in this person's life? Have they become disoriented from God? Have they opened themselves up to other things? But are there other things that are going on that are coupled together? You know, if the person comes in good faith to the Church, I think we have a responsibility to try and help them. In a sense, it's a matter of referring, too. Is it a mental health issue? A physical issue? You know, we're not trained as clinicians. We get them to the help they need.[28]

We see a connection here to what "Fr. Barron" called "spiritual health". It is the pastoral responsibility of a priest-exorcist to not only discern and properly diagnose those that seek care but to "treat the whole person", to foster their relationship with the divinity. It is for this reason that priests will often call upon saints, angels, and other heavenly residents not just to aid them in battle against supernatural evil but to facilitate a relationship between the afflicted and these holy beings. As Fr. Piero Catalano, exorcist-disciple of the late Gabriele Amorth, said in 2018:

> Which saint do I invoke most often? I have a special love for Saint Pio of Pietrelcina, who often makes himself present during exorcisms. The possessed person becomes afraid. He'll say, "The one with the beard is here!" And I reply, "By any chance, is he named Saint Pio of Pietrelcina?" The demon will respond, "No, his name is Francesco Forgione". The devil fears even to name him (Del Guercio 2018).

Recall the aforementioned diagnostic procedures of "Fr. Barron" involving blessed medals, salts, and holy water—any of which could be employed long before an exorcism takes place. Gabriele Amorth, one of the founding priests of the International Association of Exorcists, was then more of a gunslinger, using pieces of the rite as a diagnostic to see who manifested based on its reading. As "Fr. Barron" shares:

> Amorth is famously quoted as saying, "the best way you know somebody is possessed is when there's a manifestation as you pray over them". That's the Italian model. And another part of that is a willingness to go well beyond the prayers of the ritual. The American experience is very different. When somebody

> comes to me, I will spend as long as it takes. It might be six months, maybe a
> year prior to doing any exorcism. And during that time, I will speak to them at
> length. I will have them visit a medical doctor to do a complete check on them.
> I'll ask them to see a psychologist, to do a mental evaluation. I will work with
> that person and these reports to see what in their experience is answered by what
> the doctors found.[29]

It is likely this quality in Rome's most famous exorcist that garnered the admiration of
Bob Larson, an American Evangelical exorcist known for his combative persona displayed
whenever confronting a demon(iac) (see Larson (2016)). It is easy to make the joke that
the spiritual landscape of Italy is treated like that of a Spaghetti Western. As Amorth said
in 2004, "I've never been afraid of the devil. In fact, I can say he is often scared of me"
(Wilkinson 2004).

In my interview with "Fr. Drexel", he noted that the revised rite includes further
restrictions on its use—as if to tone down Amorth's thirst for battle.

> **Chavez:** Did you ever encounter either your predecessor or other exorcists using
> the ritual as a diagnostic? To see if the demoniac manifests?
>
> **"Drexel":** Certainly. My sense is that it was more readily possible to do that with
> the ancient rite. But it's restricted from doing that in the revised rite.
>
> **Chavez:** So by restricted, do you mean the instructions?
>
> **"Drexel":** As far as being able to use parts of it.
>
> **Chavez:** Because of the change in language? They took out "you dirty serpent!"
> or whatever type of diatribes are in there?
>
> **"Drexel":** Correct. So it seems like there was more freedom, shall I say, to do that
> using the ancient rite as compared to the revised rite. And, you know, I think
> there is a wisdom to being able to [provoke]. Again, it's used in a controlled
> situation. If you're like, "Okay, what's going on there?" in the sense that the
> demonic presence is elusive and is not revealing itself, you have a gut feeling it's
> there, but it's not going to show its head, then you use the ancient rite as a type
> of diagnostic tool. I think, certainly, it is done. Because what's key and different
> between the ancient and the revised rite is the whole notion of moral certitude.
> See, that's specifically mentioned in the revised rite, that before proceeding with
> the rite of exorcism, I think it's like paragraph number sixteen in the *praenotanda*,
> is that you have to have moral certitude. Where here [in the 1614 rite], there is
> not that clear distinction. And so, to be able to use it as a diagnostic tool, you
> have that freedom.[30]

This answer from "Fr. Drexel" seems to be twofold: the use of the 1614 rite as a diagnostic
for spiritual affliction is due, first, to the harsher (and, therefore, effective) "deprecative"
and "imperative" formulas that were revised in 1998, and due, second, to restricting the
"conditions for performing a major exorcism". As the *praenotanda* reads: "An Exorcist
therefore should not proceed to celebrate an Exorcism unless he has ascertained, with
moral certitude, that the one to be exorcised is really possessed by a demon and, if it is
possible, celebrate it with the consent of that person" (CDWDS 2013, p. 9).

Regarding the power of Latin and the 1614 rite, "Fr. Drexel", similar to Maginot, then
chooses not to completely disregard the revised version and its English translation.

> **"Drexel":** I use the rite I am most familiar with, which would be this one [from
> 1614], because that's what I learned. When I started in the ministry from my
> predecessor, there was no revised rite at that time. And working with the exorcist
> that was before me, [the old rite] was all he knew. Because the revision, you know,
> took place in '98. And then there were subsequent editions with amendments or
> emendations. So, I have used both.
>
> **Chavez:** Do you have a preference?

> **"Drexel":** I mean, I look at it as a sense of, you have a toolbox. And so, it depends on what the needs are. I think, in some ways, we kind of back ourselves into a corner and say, "Well, this is the *official* [rite], this is *the* one". But the thing is the circumstances are always different. Each person, each situation, how the demonic engaged someone. You know? What were the doorways? What rights it's claimed—a whole host of other things. I'm a firm believer that you don't take a bazooka to a stick fight.[31]

Thus, the specific rite deployed in a Catholic exorcism session varies according to the displayed strength of the possessing entity, the ritual familiarity of the practitioner, the cultural reception to the liturgical language of Latin, and the perceived power of its authoritative and invective rhetorical formulas.

As evident by his comment regarding the exorcist "toolbox", I should highlight that "Fr. Drexel" was also a big fan of horror films and exorcism cinema. This became apparent in our discussion of the diagnostic procedures specifically utilized by Fr. Maginot and his associates.

> **Chavez:** Fr. Mike let me go and observe his team during an early diagnostic test where they played an audio recording of the Book of Job and read the Apostles' Creed and the Our Father in Latin to see if either would provoke a demonic manifestation. Aside from the ritual as a diagnostic tool, have you also found certain things helpful? Either certain prayers, certain pieces of scripture, or Latin phrases, that help draw out the demons?
>
> **"Drexel":** Yeah, certainly, I mean, all of those things would be true. I mean, using sacramentals themselves, blessed objects, holy water, blessed salts . . .
>
> **Chavez:** Or performing in a church?
>
> **"Drexel":** Sacred places! Prayers, litanies. The prologue of John's Gospel is the most ancient text used in these rites. It's been consistently used from the beginning.
>
> **Chavez:** Really, more than Psalms?
>
> **"Drexel":** Certainly, the Psalms, but as far as New Testament scripture, consistently it's the first chapter of John's Gospel. But, you know, all of those things, sometimes it's relics of the Saints, blessed candles. It can be a whole host of things. You never know what's gonna be the trigger.
>
> **Chavez:** This toolbox often reminds me of a Van Helsing vampire hunting toolkit. The garlic and stuff. And a lot of those actually have Christian symbolism too: the hawthorn bush, the cross, holy water.
>
> **"Drexel":** Absolutely, and it's kind of all part of that. You know, Lankester Merrin [from *The Exorcist*] walks in with the satchel. But also, too, it does help us by way of reflection in another direction, of the things we've lost or forgotten about, in a sense of the sanctity, the value, the role that all of these things play as part of the rich treasure we have.[32]

It is important as scholars of religion never to lose track of people's relationships to non-human entities. For this reason, I have found it useful to employ Bruno Latour's "actor-network" theory (ANT) in this study of exorcism. For Latour (2005, p. 30), "social aggregates" could denote individuals, political organizations, cultural institutions, public policies, and so forth. This article is then concerned with a host of non-human agents ("those with agency", i.e., supernatural predators and pests, objects, places, etc.) caught within a web of dynamic relationships. In a world of culturally postulated superhuman beings, placebos, and fields of discourse, no one knows how many actants are simultaneously at work in any given situation. Actant is the theory's way of denoting a node or intersection in an actor-network, "a conglomerate of many surprising sets of agencies that have to be slowly disentangled" (p. 44). "To put it simply", Latour writes, "a good ANT account

is a narrative or a description or a proposition where all the actors do something and don't just sit there" (p. 128). The examples of non-human agents listed above "have to be actors and not simply the hapless bearers of symbolic projection" (p. 10). The exorcist, in particular, is often a moving target with "a vast array of entities swarming around it" (p. 46): their clothes, their tools, the words they speak, their assistants (of whatever realm), the tradition they represent, the families who may have consulted them, and the like. All of these social actants must be accounted for in any study of exorcism. Latour (p. 43) leads us to ask: "When an exorcist acts, who else is acting? How many agents are also present?" The priest-exorcist "toolbox" then includes the various iterations of the rite, the litany of saints, recitation of scripture, the laying on of hands, symbols of faith, the language of Latin, and even further actants (such as Padre Pio) that demonstrate an institutionalized return to the "rich treasure" of the Roman Catholic tradition. Proximity to sacrality is the power of Catholic exorcism.

Though at first glance it is easy to assume that the Catholic Church seemingly does not allow much room for ritual improvisation when reading the Roman rite of exorcism, the guidelines offered to priests in the rite's introduction (*praenotanda*) list only a few ritual customs as instructions, i.e., making signs of the cross, sprinkling holy water, and the commingling of salt (CDWDS 2013, p. 10). As Gary Thomas explains, the rest is left to the discretion of the performing priest.

> **Chavez:** What freedom do you have as an exorcist to navigate the Roman Ritual?
>
> **Thomas:** There really is no orthodox solemn rite. The rite can be subdivided into the litany of saints, the baptismal promises, the scripture reading, the psalms, the deprecatory prayers (prayers addressed to God), the imperative prayers (prayers addressed to Satan), then there's a closing prayer. That's sort of the truncated version of the rite. And that doesn't include all the preparation and stuff we do beforehand that's not part of the rite. Sometimes, the person can start manifesting at the name of a certain saint. For some reason, certain demons have a weakness toward certain saints. [ ... ] For instance, there is such a thing as an incubus spirit and a succubus spirit. Well, when you pray the title "Mary Magdalene" ("Santa Maria Magdalena"), you might get a huge reaction. You might not get any reaction to anything else, but you'll get a huge reaction to her.
>
> **Chavez:** Let's say this happens in the middle of reading the rite. Do you have the ability to stay and keep reading those sections, again and again, before moving to the next bit?
>
> **Thomas:** Yes, no one has ever told me I don't. So it wouldn't be anything to say "Santa Maria Magdalena" ten times and then move on. It's the same thing with Padre Pio, John Paul II, John Vianney, or Blessed Bartolo Longo (who was a Satanist, then had a huge conversion experience). Any of them can trigger a manifestation. We try to pray the whole rite at once but, sometimes, we'll have to stop because the person is on the floor or they're throwing up. [ ... ] But you always finish the rite then the session ends.
>
> **Chavez:** And you can linger or hang around the parts that trigger most?
>
> **Thomas:** You can deviate a bit. When you've completed the rite, you don't have to pray the whole thing all over again. You could go back and pray just the deprecatory and imperative prayers again—which I've done many times. [ ... ] In both our deliverance and exorcism sessions, we also invite Christ and the Blessed Mother in. And I have people on my team who can discern if there's demons, or human spirits, or other kinds of entities, and then we have Mary and Jesus cast them. I cast them too, and they take them to the cross. Or if it's in the case of a disembodied soul who hasn't gone for judgment, Mary will take them to Jesus for judgment.

> **Chavez:** So another form of ritual improv on the part of the priest is who to invite in the room and also what sort of Church icons are used to trigger the person.
>
> **Thomas:** Yes. Normally, I wear my blacks with the purple stole, which is the liturgical sign of my priesthood. I have a crucifix. I have holy water. I have a few relics, the team has a few relics. And we'll place them, sometimes, on the person's head.[33]

Latour's "actor-network" theory slowly disentangles the conglomerate of agencies present during an exorcism (actants such as an incubus spirit, Mary Magdalene, the relics of saints, priestly vestments, the language of Latin, and the like). As such, we achieve a greater socio-religious understanding of this modern phenomenon, specifically regarding the ritual mechanics that govern the practice of this Catholic ritual and, more generally, the religious worldviews that cohere and permit such a collision of cultural artifacts (how practitioners interpret and influence the inclusion of each of these pieces). Much of this article has been dedicated to the ways that contemporary exorcists demonstrate dissent and agency over their practices, their individual performance of the ritual and the issues that stem from the networks to which they belong.

Through these interviews, we see how the work of a priest-exorcist relies on an incorporation of the Church's powerful tradition against a range of supernatural threats as well as clear methods of discernment. Additionally, my ethnography provides an opportunity for scholars to examine a vernacular religious dimension of American Catholicism, that of the paranormal. The current era of Catholic exorcism practice is partially defined by the changing belief systems of those who request an exorcism and those who perform them. "Fr. Drexel", for instance, reveals the need for the development of new theologies within Roman Catholicism with respect to the things that "go bump in the night".

> **Chavez:** There's no consensus yet of whether there are ghosts or demons posing as ghosts.
>
> **"Drexel":** Correct. [ . . . ]
>
> **Chavez:** So where are you on the ghost issue?
>
> **"Drexel":** What I think is our theology . . . It's not well programmed out—if you will. Our theology is lacking in some ways, because we certainly have experiences. I believe that the ghost is, in a sense, the soul of a person who has died, who is no longer bound by physicality, who is stuck, is not moving forward into eternity, remains attached to a place. I believe that can happen. Separately, a demon, masquerading as [a ghost]. I think that can happen as well. I mean, fundamentally, you are always dealing with a liar when it comes to the demonic. But, because we've got a long history with the tradition of the Church, where you have these manifestations people encounter, even classic stories of, you know, a priest who has died who didn't properly care for mass intentions and things like that. I heard one priest, a very credible exorcist, talking at length about, you know, walking in and having to pray while he was training as an exorcist. This was, I believe, in the New York area or out east, anyway. And, you know, this odd thing is happening in this rectory after this elderly priest died, and the whole situation, the long and short of it, is that he had a whole drawer full of mass intentions, that weren't celebrated, and that had been sitting there, and he could not rest, and, you know, for whatever reason, God permitted him to make things right or allow that process to take place. And it clearly wasn't demonic. I think, you know, the problem is, we don't have neat categories to fit all these concepts.[34]

As one can see, the practice of a modern American exorcist, similar to its enmeshment with sensationalist media and popular entertainment, is thoroughly embedded within a culture of Catholic folklore. Exorcists hear and share collections of stories involving unsuspecting Catholic women that purchase dangerous occult objects at New Age bookstores; or stories

of other priests neglecting a whole drawer of mass intentions and, though now dead, unable to transcend this world until such unfinished business was resolved.

Yet, within these priestly folktales, one also observes a call for institutional expansion of the *supernatural* or *preternatural* categories of Catholic theology. Of all the ways the Roman Catholic Church seeks to "modernize" its office of exorcist, few seem to have foreseen a need for it to reevaluate its ability to explain the daily experiences of its members. Such a call was also observed in my interviews with "Fr. Barron", himself sharing a priestly folktale (seemingly imagined though still representative) of the types of abilities and strange occurrences experienced at the vernacular religious level.

> **"Barron":** I don't know if you come across this but there is no one that I know of in the Church who's doing a theology or a theological investigation of all of this stuff. [ . . . ] For example, magic. You know, does magic work?
>
> **Chavez:** I always found that curious. If you're afraid of somebody picking up the Egyptian Book of the Dead, or something akin, and using it for magic, you're giving credence to its power and to its possibility—which seems odd to me.
>
> **"Barron":** I don't know. That's a question that I would be willing to investigate. My sense is that those things happen and have an effectiveness.
>
> **Chavez:** Even though they don't come from God?
>
> **"Barron":** Yeah, they come from the Enemy. But I would say, and I'm not going to argue this or put my life on it, but you have in the world the *supernatural* and the *natural*. But then I think you also have the *preternatural*. And at the level of the preternatural, you have many gifts and gray areas.
>
> **Chavez:** And that part isn't really being worked out right now?
>
> **"Barron":** Exactly. If you have, for example, a grandmother who wakes up one morning and says, "My grandson was killed in a crash last night. I saw it happen. And I know he's dead" and then, 10 minutes later, the doorbell rings and police are at the door and, sure enough, he got killed. How does that happen? Well, to my mind, there is a preternatural area that is not supernatural and not natural; it is in between. And it is at that level, I think, that magic can be located. Now, magic, it's not a trick but something that can be manipulated, the desire to use the preternatural for my benefit. But I also think that the preternatural can be used not just for people to know things. I mean, I bet you everybody alive knows somebody who says, "Oh, go answer the phone. It's about to ring. It's Aunt Jan". And you say, "huh?" and then the phone rings. "Oh, it's Aunt Jan". That happens so often that people can't deny that.[35]

Though this story from "Fr. Barron" concerning a preternatural grandmother is presented parabolically or hypothetically, it should be noted that such a vernacular religious dimension encompasses far more than religious worldviews regarding Satanic conspirators, supernatural predators, and possession experiences. However, when asked about his religious associations with Charismatic Catholicism, "Fr. Barron" clarified that his view of a preternatural cosmology functions independently of the Neo-Pentecostal tradition.

> **Chavez:** Do you consider yourself a Charismatic?
>
> **"Barron":** No, I don't think it has anything to do with charisms or Charismatics. I think this is simply, you know, that there are people who have a giftedness, some call it "second sight", where they simply know things. They're sensitive. [ . . . ] We don't know how that works. And that needs to be explored because it is the realm where the natural and what we would call "the higher faculties" [intersect]. Do you know what bilocation is? Padre Pio is often talked about as being in two places at once. How does that happen? Well, we Catholics, and the tradition, just say, "That's a miracle!" But how is it a miracle? And if we believe that Padre Pio can be in two places at once, why would we not believe that somebody's

personality could be split off and part of it travel? We don't have a theological explanation for ["bilocation", "dissociation", or "astral projection"].

**Chavez:** There's a lot of miracles attributed to saints that would tap into this preternatural realm you speak of.

**"Barron":** Yeah. The fact that our Lord allows it to happen is the miracle. [But there's more to it than that]. There's a priest in Mexico, Father Rogelio Alcantara. He lives in Mexico City and he, my bishop, and I sat for a week a year ago and went over these questions. And said, for example, "How is it possible that I curse something and you are affected by the curse?" "How is it possible that I curse this and give it to you and you take it and you receive a curse?" So we sat there and talked for days and days about . . . "What is involved?" "What is a curse?" And it always involves this preternatural level and he's the only guy I know who's developing or working on this and he's only doing it because I keep pestering him with these issues.

In my research of the intersection between the *demonic* and *paranormal* (given that categories such as *preternatural* are far less studied), I stumbled upon a translation of an exorcism rite, entitled *Exorcismus domus a dæmonio vexate* ("Exorcism of a house troubled with an evil spirit"), included as an appendix within the 1631 edition of the *Rituale Romanum*. This rite was translated into English ("Exorcism of Haunted Houses") by Herbert Thurston and included as an appendix within his Ghosts and Poltergeists (1953) monograph. When I showed this document to "Fr. Barron", he expressed familiarity with the rite while also providing the following annotations:

First of all, there is in the Roman Ritual a prayer specifically for the exorcism of a place. The prayers are relatively late, dating only from the pontificate of Pope Leo XIII, but is now very widely used by exorcists. It is also sometimes used as a diagnostic tool. Secondly, this blessing/exorcism (Exorcismus domus a dæmonio vexate) is well known to me and to other exorcists as well. However, in my experience, there is one significant difference. This extract calls for three trips all through the house, each time blessing the house with holy water. In my experience, the house is blessed three separate times, but the first time with water, the second time with incense, and the third time with sacred oil, which is used to trace the cross upon the walls and doors of the house. Windows and mirrors are always anointed in this same way. Also, there are specific prayers that close each of the perambulations and the recitation of each of the three sets of Gradual Psalms.[36]

In my 2020 follow-up interview with Gary Thomas, he also mentioned the use of the Pope Leo XIII Prayer when exorcising a house—which he prays in English. Interestingly, in the excerpt below, Thomas also shares further accounts of ritual improvisation due to the 2020 social distancing protocols.

**Thomas:** I go to a lot of houses. Since COVID, I've gone to a lot of houses. And the ritual takes about maybe 15 minutes to pray. What I've done with COVID is a little different than pre-COVID. In the past, I would do most of the rite at the main doorway then I would sprinkle every door in the house. I'd pray the Hail Mary in every room with the family. And then we would go into the backyard and the front yard where I would use exorcise water with exorcise salt then pray the prayer throughout the whole property that's claimed by that family. What I do now is I pray most of the ritual outside at the front door and then I go in with a mask and go to all the rooms and do exactly what I just told you before. But it's much quicker.

**Chavez:** The latter minimizes the time that you're spending in the house.[37]

As I mentioned before, this article on American exorcism intersects a number of competing narratives. Thus far, we have seen a rise in institutional support of an office

previously disregarded by the Church, various methods of modernization including the appropriation of medical and psychological discourse and the use of popular media to promote a public awareness on issues related to supernatural evil and Satanic practices, as well as the ritual mechanics involved in each exorcism that reportedly grant the ritual its efficacy. However, this call for the Church to reformulate its theological categories is significant as it bears an imperative quality, not just descriptive. These priest-exorcists signal the path by which Roman Catholic tradition can continue to offer its subjects a guiding hand when processing their daily experiences of terror, dread, and the like. Fr. Thomas, for instance, reports a host of supernatural predators encountered throughout his exorcism ministry, including demons, incubi/succubi, fallen angels, disembodied souls, but also aliens and interdimensional beings.

> We've . . . had a few encounters with paranormal entities. Aliens? We've had a few. Now, the reason that we've come to that conclusion is because (no. 1) they don't know who Jesus Christ is. But when you ask them "Do you know Yahweh?", there's a huge reaction! Now John Paul II, when he was still pope, was asked at least once if he believed in the existence of life on other planets. And he said, "If there is life on other planets, God is still sovereign over all". So when we were getting this person in a trance and getting no kind of reaction from the names of Jesus Christ or the Blessed Mother, somebody on my team said, "Maybe this is an entity from somewhere else". We would say an "entity of the cosmos", because there are entities that are related to the elements (fire, water, wind, and earth). They may not be demons, but they are spirits, separate from the demons attached to many of the false gods in Buddhism and Hinduism—because we've dealt with that too. But the aliens (no. 2) are also not in league with Satan. Some are. And, I mean, just in the last two months there's been stuff now that's finally come out from the Department of Defense, reports of, you know, "unidentified flying objects". I think the government has concealed this for a long time. And I do believe that there is life on other planets. I also think there is life in other dimensions. I know because I've had several people come to me who have used drugs (like ayahuasca) to interact with these entities.[38]

As Bader et al. (2017, p. 15) propose, "Paranormalism generally lacks the stability and organization that characterize successful religious groups, operating on the periphery of American religion, spreading through conferences, the media, and the Internet rather than through sermons and encyclicals". The paranormal then belongs to the realm of folklore (however it may be distributed)—that which is ultimately, according to Barre Toelken (1996, p. 34), "local, communal, and informal". The only consistent definition of the subject is that of the "ideas and experiences that have not yet been adopted, at least wholly, by the dominant religions in a given society". Experiences of demons and angels, for instance, can easily be explained by scripture, centuries of theological intricacies, and popular religious literature and are, thus, welcomed into the dogmatic territory of the Catholic Church. Far less direction, however, is offered to those that claim encounters with aliens or interdimensional beings. The statistical data presented in Bader et al. (2017) seem to suggest that those who have moderate levels of religiosity are then most likely to report a paranormal experience. It is within this distinct demographic that a "bridge" is formed between the worlds of religious and paranormal intrigue.

> Consider . . . the diverse collection of topics and subjects that appear in the paranormal section of bookstores. Scattered among the UFO books, astrology training manuals, and collections of ghost tales you are likely to find books that discuss sightings of the Virgin Mary, stories of demons, possessions, and exorcism, as well as collected tales of claimed miraculous healings and rescues by guardian angels. [ . . . ] Books on the same topics will also be found in most Christian bookstores and in the Christian section of general bookstores. An exorcism tale can carry different meanings depending upon the type of book in which it

appears. A book targeted at Christian audiences is likely to frame exorcisms as proof of the reality of Satan and a warning to Christians to get right with God and avoid the "occult". A book meant for general audiences will treat exorcisms as a fascinating and frightening mystery but will avoid explicit religious overtones or suggestions that a conversion to Christianity is a means of avoiding possession. A Christian author may write of the power of Ouija boards, but rather than focus upon it as a potential means to contact a dead relative, the object is feared as an instrument through which demons may attack the unwary. Exorcisms, Virgin Mary sightings, guardian angel tales, and the like serve as a potential bridging area between religion and the paranormal. Perhaps an interest in such topics will lead conservative Christians to develop an interest in paranormal subjects, such as UFOs and ESP, or maybe an interest in guardian angels could draw an otherwise nonreligious person into a faith (p. 194).

## 5. Conclusions

By now it should be clear that the interviewed individuals above gesture to a much larger portrait of vernacular American Catholicism than simply contemporary exorcism practice. The goal of this article has been to construct a "modern" portrait of an "archaic" ritual, detailing how contemporary Catholic exorcists are currently "modernizing" their practices and ministries. While this study serves as a hybrid extension to Cuneo (2001) and Bader et al. (2017), said union dictates still further research and data collection. Thus, through an examination of American religious folklore and ethnographic research on the practice of exorcism, we have learned that despite the popular perception that the ritual does not belong in modern society, the practice persists and, therefore, requires further analysis. Since 1998, the Catholic Church has revamped its office of exorcist with significant institutional support in the form of multiple training programs, papal mandates, revised ritual manuals, and an increased number of Catholic priest-exorcists. In their interviews, my Roman Catholic collaborators each present themselves as a trained skeptic that ultimately pursues a form of "spiritual health" for their subjects. They interpret their ritual actions as a mechanism by which to foster a relationship between the afflicted and the divinity, bringing each of them into a proximate sacrality. They openly voice dissent regarding the Church's governance of their practice and exercise agency as a practitioner to diagnose and discern as well as ritually improvise.

Of all the methods by which contemporary exorcists "modernize" their practices and ministries, the two most obvious include, first, the appropriation of medical and psychological discourse and, second, the use of popular media to promote a public awareness of both supernatural evil and Satanic practices. My study demonstrates that the American public still lives in the aftermath of the Satanic Panic, a clear period in Christian history that continues to produce not only discursive fear regarding the challenge of Satanism but institutional resources to confront it. This article further reveals a call for the Church to reformulate its theological categories to include entities and experiences typically registered in the academy as "paranormal". These priest-exorcists, first, signal the path by which Roman Catholic tradition can continue to offer its subjects a guiding hand when processing their daily experiences and, second, signal to scholars that the practice of exorcism encompasses much more than what was previously reported in academic studies. A further representative sample of contemporary exorcists would then allow us to see the concurrent sets of religious perspectives and networks that operate simultaneously to the traditional Roman Catholic priests interviewed above.

**Funding:** This research was made possible through the support of two UC Continuing Central Fellowships and the Higher Education Emergency Relief Fund, namely the Graduate Opportunity Fellowship (2019–2020), Graduate Research Mentorship Program Fellowship (2020–2021), and CARES Act MSI Summer Grant (2021).

**Institutional Review Board Statement:** Ethical review and approval were waived for this study by the University of California, Santa Barbara due to qualifying as Exempt from the Federal Regulations at 45 CFR 46.104(d) under Category 2.

**Informed Consent Statement:** Informed consent was obtained from all subjects involved in the study. Written informed consent has been obtained from the patients to publish this paper.

**Acknowledgments:** This article was commissioned by the late Leonard Primiano in Spring 2020 as a contribution to his edited volume, *Vernacular Catholicism: Folkloristic Studies of Catholic Culture*. Though I submitted this article to Leonard in Fall of the same year, because of his worsening condition, I do not believe he ever read it. I dedicate this research to Leonard's legacy—not just because of his personal support of my work but due to the overall spirit of the analysis. This research simply could not exist without Leonard's investigation of vernacular religion, "religion as it is lived". I am grateful to my collaborators for their patience, warmth, and guidance. My work was improved due to their critical insights along with the comments and suggestions offered by my advisors, colleagues, and anonymous reviewers. I am indebted specifically to Rudy V. Busto, Dwight Reynolds, Elizabeth Pérez, Maharshi Vyas, and Laurel Zwissler. I recognize also all those who have supported my study of exorcism since 2012, namely Nora Rubel, Joshua Dubler, Richard Hecht, José Cabezón, Vesna Wallace, David Gordon White, Christine Thomas, Juan Campo, and the late John Mohr. I offer a special thanks to my friend and colleague Sarah Veeck for assisting in the transcription of my interviews. Finally, I reserve the deepest of appreciation for my wife Stefany Olague. I cherish my marriage because of her kind heart and loving soul.

**Conflicts of Interest:** The author declares no conflict of interest.

## Notes

1  Giordan (2020) examines a contemporary Italian exorcist "protocol" for discerning "true" cases of demonic possession. Similar to Suhr (2019), which documents a Palestinian refugee's experience of jinn possession in Denmark and his dual forms of treatment, "both exorcist and psychiatrist", Giordan argues, ultimately "go through procedures which legitimize and reinforce each other" (p. 95). Sax (2020) demonstrates the same.

2  "The Roman Catholic rite of exorcism, so effectively dramatized in Blatty's movie, was first formally laid down in the *Rituale Romanum* of 1614, and the 1917 Code of Canon Law, which was in effect until 1983, mandated that every bishop officially appoint an exorcist for his diocese. (As we've seen, however, this wasn't always followed through in places such as the United States.)" (Cuneo 2001, p. 130).

3  For information on the inception of the course, see Baglio (2009), pp. 7, 16–17.

4  This overview of "the church's expanding corps of exorcists", as mobilized by a developing "discourse of evil", is similarly covered in Csordas (2017, p. 294). "The trajectory of this development . . . suggests that exorcism be understood not only as a thriving form of religious practice but also as a dynamic social phenomenon".

5  Interview with "Fr. Drexel" (in-person), 9 December 2019.

6  Interview with Fr. Gary Thomas in Saratoga, CA, 8 June 2018.

7  To be clear, this does not translate to 150 or 175 dioceses with appointed exorcists.

8  See Cuneo (2001, p. 5) for the scholar's contextualization of Catholic priesthood amidst the recent controversies and scandals: It is the area of exorcism where "the priest-as-exorcist . . . has somehow managed, in defiance of all odds, to retain a heroic grip on the popular American imagination".

9  For a more detailed biography on Thomas, see Baglio (2009), pp. 22–29.

10  Interview with "Fr. Barron" (in-person), 13 December 2019.

11  Interview with "Fr. Barron" (in-person), 13 December 2019.

12  Interview with "Fr. Barron" (email), 9 October 2019.

13  Interview with "Fr. Barron" (in-person), 13 December 2019.

14  See Maginot (2020) for the published form of the letter.

15  Interview with Fr. Michael Maginot in Merrillville, IN, 6 December 2019.

16  For more on Amorth, see Giordan and Possamai (2018) and Young (2016).

17  For a general overview, see Frankfurter (2006) and Cuneo (2001), pp. 195–216.

18  See Frankfurter (2006), pp. 68–69 and McCloud (2015), pp. 6–10, 21 for a discussion of the influx of practitioners following the moral panic of the 1980s. Following his interviews with local Vatican priests, Csordas (2017, p. 294) cites a more recent Italian moral panic as the impetus for the Catholic Church's renewed interest in Satanism and subsequent recruitment and training of more exorcists. "In the wake of a satanism scare in Italy following a series of murders in 1998 and 2004 involving the heavy metal

rock band Beasts of Satan, word spread that the Vatican wanted every diocese to appoint an exorcist, and a training course for exorcists launched at a pontifical university in Rome has been conducted annually since 2005".

[19]  These evil conspirators are said to gravitate towards ritual perversion and sexual deviation, i.e., blood-sacrifice, cannibalism, infant-murder, orgies, and child pornography. See Frankfurter (2006) and Lanning (1992, p. 22).

[20]  Cf. Table 3.4 ("Reasons for the Visit") in Giordan and Possamai (2018, p. 55).

[21]  Interview with Fr. Gary Thomas in Saratoga, CA, 8 June 2018.

[22]  Interview with Fr. Michael Maginot in Merrillville, IN, 6 December 2019.

[23]  Interview with Fr. Gary Thomas in Saratoga, CA, 8 June 2018.

[24]  Interview with "Fr. Barron" (phone), 4 September 2020.

[25]  Interview with Fr. Michael Maginot in Merrillville, IN, 6 December 2019.

[26]  Interview with Fr. Gary Thomas (Zoom), 27 September 2020.

[27]  Interview with Fr. Gary Thomas in Saratoga, CA, 8 June 2018.

[28]  Interview with "Fr. Drexel" (in-person), 9 December 2019.

[29]  Interview with "Fr. Barron" (in-person), 13 December 2019.

[30]  Interview with "Fr. Drexel" (in-person), 9 December 2019.

[31]  Interview with "Fr. Drexel" (in-person), 9 December 2019.

[32]  Interview with "Fr. Drexel" (in-person), 9 December 2019.

[33]  Interview with Fr. Gary Thomas (Zoom), 27 September 2020.

[34]  Interview with "Fr. Drexel" (in-person), 9 December 2019.

[35]  Interview with "Fr. Barron" (in-person), 13 December 2019.

[36]  Interview with "Fr. Barron" (email), 16 October 2019.

[37]  Interview with Fr. Gary Thomas (Zoom), 27 September 2020.

[38]  Interview with Fr. Gary Thomas (Zoom), 27 September 2020.

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
