# Peer review of "Modern Practice, Archaic Ritual: Catholic Exorcism in America"

_religions, doi:10.3390/rel12100811_

Round 1
Reviewer 1 Report
This manuscript contains very rich ethnographic data. The entire article is structured around the contribution of the 4 interviewees. The topic is intriguing and is well elaborated. I would suggest only two minor additions. When the author mentions vernacular religious practices, I think the source should be given. For a folklorist reader it is obvious that this term comes from Dr. Primiano, but for the general religious studies readership it should be made clear why 'vernacular' is chosen over other lived/everyday religion approaches. Also, a reference would be nice to some of the works of Thomas Csordas (Possession and Psychopathology, Faith and Reason; The Sacred Self) as he is one of the main experts in the field of exorcism and Catholicism in the US.
Author Response
Dear Anonymous Reviewer 1,
Thank you for seeing considerable promise in my recent submission. I appreciate your words of praise.
You will find that I have included references to Leonard Primiano (1995) and Thomas Csordas (2017) not just as endnotes but within the body of the manuscript. Primiano’s influence has been addressed in my section on Methodologies, lines 101-113, while Csordas is cited in both lines 56-61 and endnotes 4 and 19.
I wish to convey also a special sense of gratitude for the Csordas suggestion in that his concerns appear similar to my own.
Best,
William Chavez
Reviewer 2 Report
This is a most interesting contribution to an underrated field of research. The author is clearly highly competent and well-informed. I do not see any need for major revisions. I do not quite see why the exorcists in question are not clearly identified.
Author Response
Dear Anonymous Reviewer 2,
Thank you for seeing considerable promise in my recent submission.
In regards your concerns, I offer the following. As per the confidentiality agreement included in my IRB consent forms, I honor the anonymity requested by some of my collaborators. My understanding is that a Catholic priest-exorcist known to the general public is an unsavory position due to the exorbitant amount of calls and requests made from those seeking an exorcism and those associated with popular media. I reference as much in lines 141-144 of the manuscript.
Thank you for thanking the time to review my manuscript. I appreciate your words of praise and am excited to soon be published within this journal.
Best,
William Chavez